# OPEN LOOP HYPERPARAMETER OPTIMIZATION AND DETERMINANTAL POINT PROCESSES

## ABSTRACT

Driven by the need for parallelizable hyperparameter optimization methods, this paper studies *open loop* search methods: sequences that are predetermined and can be generated before a single configuration is evaluated. Examples include grid search, uniform random search, low discrepancy sequences, and other sampling distributions. In particular, we propose the use of $k$-determinantal point processes in hyperparameter optimization via random search. Compared to conventional uniform random search where hyperparameter settings are sampled independently, a $k$-DPP promotes diversity. We describe an approach that transforms hyperparameter search spaces for efficient use with a $k$-DPP. In addition, we introduce a novel Metropolis-Hastings algorithm which can sample from $k$-DPPs defined over any space from which uniform samples can be drawn, including spaces with a mixture of discrete and continuous dimensions or tree structure. Our experiments show significant benefits in realistic scenarios with a limited budget for training supervised learners, whether in serial or parallel.

## 1 INTRODUCTION

Hyperparameter values—regularization strength, model family choices like depth of a neural network or which nonlinear functions to use, procedural elements like dropout rates, stochastic gradient descent step sizes, and data preprocessing choices—can make the difference between a successful application of machine learning and a wasted effort. To search among many hyperparameter values requires repeated execution of often-expensive learning algorithms, creating a major obstacle for practitioners and researchers alike.

In general, on iteration (evaluation) $k$, a hyperparameter searcher suggests a $d$-dimensional hyperparameter configuration $x_k \in \mathcal{X}$ (e.g., $\mathcal{X} = \mathbb{R}^d$ but could also include discrete dimensions), a worker trains a model using $x_k$, and returns a validation loss of $y_k \in \mathbb{R}$ computed on a hold out set. In this work we say a hyperparameter searcher is **open loop** if $x_k$ depends only on $\{x_i\}_{i=1}^{k-1}$; examples include choosing $x_k$ uniformly at random (Bergstra & Bengio, 2012), or $x_k$ coming from a low-discrepancy sequence (c.f., Iacò (2015)). We say a searcher is **closed loop** if $x_k$ depends on both the past configurations and validation losses $\{(x_i, y_i)\}_{i=1}^{k-1}$; examples include Bayesian optimization (Snoek et al., 2012) and reinforcement learning methods (Zoph & Le, 2016). Note that open loop methods can draw an infinite sequence of configurations before training a single model, whereas closed loop methods rely on validation loss feedback in order to make suggestions.

While sophisticated closed loop selection methods have been shown to empirically identify good hyperparameter configurations faster (i.e., with fewer iterations) than open loop methods like random search, **two trends have rekindled interest in embarrassingly parallel open loop methods**: 1) modern deep learning model are taking longer to train, sometimes up to days or weeks, and 2) the rise of cloud resources available to anyone that charge not by the number of machines, but by the number of CPU-hours used so that 10 machines for 100 hours costs the same as 1000 machines for 1 hour.

This paper explores the landscape of open loop methods, identifying tradeoffs that are rarely considered, if at all acknowledged. While random search is arguably the most popular open loop method and chooses each $x_k$ independently of $\{x_i\}_{i=1}^{k-1}$, it is by no means the only choice. In many ways uniform random search is the least interesting of the methods we will discuss because we will advocate for methods where $x_k$ depends on $\{x_i\}_{i=1}^{k-1}$ to promote **diversity**. In particular, we will focus on

drawing $\{x_i\}_{i=1}^k$ from a $k$-**determinantal point process (DPP)** (Kulesza et al., 2012). We introduce a sampling algorithm which allows DPPs to support real, integer, and categorical dimensions, any of which may have a tree structure, and we describe connections between DPPs and Gaussian processes (GPs).

In synthetic experiments, we find our diversity-promoting open-loop method outperforms other open loop methods. In practical hyperparameter optimization experiments, we find that it significantly outperforms other approaches in cases where the hyperparameter values have a large effect on performance. Finally, we compare against a closed loop Bayesian optimization method, and find that sequential Bayesian optimization takes, on average, more than ten times as long to find a good result, for a gain of only 0.15 percent accuracy on a particular hyperparameter optimization task.

Open source implementations of both our hyperparameter optimization algorithm (as an extension to the hyperopt package (Bergstra et al., 2013)) and the MCMC algorithm introduced in Algorithm 2 are available.[1]

## 2 RELATED WORK

While this work focuses on open loop methods, the vast majority of recent work on hyperparameter tuning has been on closed loop methods, which we briefly review.

### 2.1 CLOSED LOOP METHODS

Much attention has been paid to sequential model-based optimization techniques such as Bayesian optimization (Bergstra et al., 2011; Snoek et al., 2012) which sample hyperparameter spaces adaptively. These techniques first choose a point in the space of hyperparameters, then train and evaluate a model with the hyperparameter values represented by that point, then sample another point based on how well previous point(s) performed. When evaluations are fast, inexpensive, and it's possible to evaluate a large number of points (e.g. $k = \Omega(2^d)$ for $d$ hyperparameters) these approaches can be advantageous, but in the more common scenario where we have limited time or a limited evaluation budget, the sequential nature of closed loop methods can be cumbersome. In addition, it has been observed that many Bayesian optimization methods with a moderate number of hyperparameters, when run for $k$ iterations, can be outperformed by sampling $2k$ points uniformly at random (Li et al., 2017), indicating that even simple open loop methods can be competitive.

Parallelizing Bayesian optimization methods has proven to be nontrivial, though many agree that it's vitally important. While many algorithms exist which can sample more than one point at each iteration (Contal et al., 2013; Desautels et al., 2014; González et al., 2016; Kandasamy et al., 2018), the sequential nature of Bayesian optimization methods prevent the full parallelization open loop methods can employ. Even running two iterations (with batches of size $k/2$) will take on average twice as long as fully parallelizing the evaluations, as you can do with open loop methods like grid search, sampling uniformly, or sampling according to a DPP.

One line of research has examined the use of $k$-DPPs for optimizing hyperparameters in the context of parallelizing Bayesian optimization (Kathuria et al., 2016; Wang et al., 2017). At each iteration within one trial of Bayesian optimization, instead of drawing a single new point to evaluate from the posterior, they define a $k$-DPP over a relevance region from which they sample a diverse set of points. They found their approach to beat state-of-the-art performance on a number of hyperparameter optimization tasks, and they proved that generating batches by sampling from a $k$-DPP has better regret bounds than a number of other approaches. They show that a previous batch sampling approach which selects a batch by sequentially choosing a point which has the highest posterior variance (Contal et al., 2013) is just approximating finding the maximum probability set from a $k$-DPP (an NP-hard problem (Kulesza et al., 2012)), and they prove that sampling (as opposed to maximization) has better regret bounds for this optimization task. We use the work of Kathuria et al. (2016) as a foundation for our exploration of fully-parallel optimization methods, and thus we focus on $k$-DPP sampling as opposed to maximization.

So-called configuration evaluation methods have been shown to perform well by adaptively allocating resources to different hyperparameter settings (Swersky et al., 2014; Li et al., 2017). They initially

---

[1]Anonymized URL; will be provided on publication.

choose a set of hyperparameters to evaluate (often uniformly), then partially train a set of models for these hyperparameters. After some fixed training budget (e.g., time, or number of training examples observed), they compare the partially trained models against one another and allocate more resources to those which perform best. Eventually, these algorithms produce one (or a small number) of fully trained, high-quality models. In some sense, these approaches are orthogonal to open vs. closed loop methods, as the diversity-promoting approach we advocate can be used as a drop-in replacement to the method used to choose the initial hyperparameter assignments.

## 2.2 SAMPLING PROPORTIONAL TO THE POSTERIOR VARIANCE OF A GAUSSIAN PROCESS

GPs have long been lauded for their expressive power, and have been used extensively in the hyperparameter optimization literature. Hennig & Garnett (2016) show that drawing a sample from a $k$-DPP with kernel $\mathcal{K}$ is equivalent to sequentially sampling $k$ times proportional to the (updated) posterior variance of a GP defined with covariance kernel $\mathcal{K}$. This sequential sampling is one of the oldest hyperparameter optimization algorithms, though our work is the first to perform an in-depth analysis. Additionally, this has a nice information theoretic justification: since the entropy of a Gaussian is proportional to the log determinant of the covariance matrix, points drawn from a DPP have probability proportional to $\exp(\text{information gain})$, and the most probable set from the DPP is the set which maximizes the information gain. With our MCMC algorithm presented in Algorithm 2, we can draw samples with these appealing properties from any space for which we can draw uniform samples. The ability to draw $k$-DPP samples by sequentially sampling points proportional to the posterior variance grants us another boon: if one has a sample of size $k$ and wants a sample of size $k+1$, only a single additional point needs to be drawn, unlike with the sampling algorithms presented in Kulesza et al. (2012). Using this approach, we can draw samples up to $k = 100$ in less than a second on a machine with 32 cores.

## 2.3 OPEN LOOP METHODS

As discussed above, recent trends have renewed interest in open loop methods. While there exist many different batch BO algorithms, analyzing these in the open loop regime (when there are no results from function evaluations) is often rather simple. As there is no information with which to update the posterior mean, function evaluations are hallucinated using the prior or points are drawn only using information about the posterior variance. For example, in the open loop regime, Kandasamy et al. (2018)'s approach without hallucinated observations is equivalent to uniform sampling, and their approach with hallucinated observations (where they use the prior mean in place of a function evaluation, then update the posterior mean and variance) is equivalent to sequentially sampling according to the posterior variance (which is the same as sampling from a DPP). Similarly, open loop optimization in SMAC (Hutter et al., 2012) is equivalent to first Latin hypercube sampling to make a large set of diverse candidate points, then sampling $k$ uniformly among these points.

Recently, uniform sampling was shown to be competitive with sophisticated closed loop methods for modern hyperparameter optimization tasks like optimizing the hyperparameters of deep neural networks (Li et al., 2017), inspiring other works to explain the phenomenon (Ahmed et al., 2016). Bergstra & Bengio (2012) offer one of the most comprehensive studies of open loop methods to date, and focus attention on comparing random search and grid search. A main takeaway of the paper is that uniform random sampling is generally preferred to grid search[2] due to the frequent observation that some hyperparameters have little impact on performance, and random search promotes more diversity in the dimensions that matter. Essentially, if points are drawn uniformly at random in $d$ dimensions but only $d' < d$ dimensions are relevant, those same points are uniformly distributed (and just as diverse) in $d'$ dimensions. Grid search, on the other hand, distributes configurations aligned with the axes so if only $d' < d$ dimensions are relevant, many configurations are essentially duplicates.

However, grid search does have one favorable property that is clear in just one dimension. If $k$ points are distributed on $[0, 1]$ on a grid, the maximum spacing between points is equal to $\frac{1}{k-1}$. But if points are uniformly at random drawn on $[0, 1]$, the expected largest gap between points scales as $\frac{1}{\sqrt{k}}$. If, by

---

[2]Grid search uniformly grids $[0, 1]^d$ such that $x_k = (\frac{i_1}{m}, \frac{i_2}{m}, \dots, \frac{i_d}{m})$ is a point on the grid for $i_j = 0, 1, \dots, m$ for all $j$, with a total number of grid points equal to $(m + 1)^d$.

bad luck, the optimum islocated in this largest gap, this difference could be considerable; we attempt to quantify this idea in the next section.

## 3 MEASURES OF SPREAD

Quantifying the spread of a sequence $\mathbf{x} = (x_1, x_2, \ldots, x_k)$ (or, similarly, how well $\mathbf{x}$ covers a space) is a well-studied concept. In this section we introduce discrepancy, a quantity used by previous work, and dispersion, which we argue is more appropriate for optimization problems.

### 3.1 DISCREPANCY

Perhaps the most popular way to quantify the spread of a sequence is star discrepancy. One can interpret the star discrepancy as a multidimensional version of the Kolmogorov-Smirnov statistic between the sequence $\mathbf{x}$ and the uniform measure; intuitively, when $\mathbf{x}$ contains points which are spread apart, star discrepancy is small. We include a formal definition in Appendix A.

Star discrepancy plays a prominent role in the numerical integration literature, as it provides a sharp bound on the numerical integration error through the the Koksma-Hlawka inequality (given in Appendix B) (Hlawka, 1961). This has led to wide adoption of low discrepancy sequences, even outside of numerical integration problems. For example, Bergstra & Bengio (2012) analyzed a number of low discrepancy sequences for some optimization tasks and found improved optimization performance over uniform sampling and grid search. Additionally, low discrepancy sequences such as the Sobol sequence[3] are used as an initialization procedure for some Bayesian optimization schemes (Snoek et al., 2012).

### 3.2 DISPERSION

Previous work on open loop hyperparameter optimization focused on low discrepancy sequences (Bergstra & Bengio, 2012; Bousquet et al., 2017), but optimization performance—how close a point in our sequence is to the true, fixed optimum—is our goal, not a sequence with low discrepancy. As discrepancy doesn't directly bound optimization error, we turn instead to dispersion

$$d_k(\mathbf{x}) = \sup_{x \in [0,1]^d} \min_{1 \leq i \leq k} \rho(x, x_i),$$

where $\rho$ is a distance (in our experiments $L_2$ distance). Intuitively, the dispersion of a point set is the radius of the largest Euclidean ball containing no points; dispersion measures the worst a point set could be at finding the optimum of a space.

Following Niederreiter (1992), we can bound the optimization error as follows. Let $f$ be the function we are aiming to optimize (maximize) with domain $\mathcal{B}$, $m(f) = \sup_{x \in \mathcal{B}} f(x)$ be the global optimum of the function, and $m_k(f; \mathbf{x}) = \sup_{1 \leq i \leq k} f(x_i)$ be the best-found optimum from the set $\mathbf{x}$. Assuming $f$ is continuous (at least near the global optimum), the modulus of continuity is defined as

$$\omega(f; t) = \sup_{\substack{x,y \in \mathcal{B} \\ \rho(x,y) \leq t}} |f(x) - f(y)|, \text{ for some } t \geq 0.$$

**Theorem 1.** *(Niederreiter, 1992) For any point set* $\mathbf{x}$ *with dispersion* $d_k(\mathbf{x})$, *the optimization error is bounded as*

$$m(f) - m_k(f; \mathbf{x}) \leq \omega(f; d_k(\mathbf{x})).$$

Dispersion can be computed efficiently (unlike discrepancy, $D_k(\mathbf{x})$, which is NP-hard (Zhigljavsky & Zilinskas, 2007)), and we give an algorithm in Appendix C. Dispersion is at least $\Omega(k^{-1/d})$, and while low discrepancy implies low dispersion ($d^{-1/2} d_k(\mathbf{x}) \leq \frac{1}{2} D_k(\mathbf{x})^{1/d}$), the other direction does

---

[3]Bergstra & Bengio (2012) found that the Niederreiter and Halton sequences performed similarly to the Sobol sequence, and that the Sobol sequence outperformed Latin hypercube sampling. Thus, our experiments include the Sobol sequence (with the Cranley-Patterson rotation) as a representative low-discrepancy sequence.

not hold.[4] Therefore we know that the low-discrepancy sequences evaluated in previous work are also low-dispersion sequences in the big-$O$ sense, but as we will see they may behave quite differently. Samples drawn uniformly are not low dispersion, as they have rate $(\ln(k)/k)^{1/d}$ (Zhigljavsky & Zilinskas, 2007).

Optimal dispersion in one dimension is found with an evenly spaced grid, but it's unknown how to get an optimal set in higher dimensions.[5] Finding a set of points with the optimal dispersion is as hard as solving the circle packing problem in geometry with $k$ equal-sized circles which are as large as possible. Dispersion is bounded from below with $d_k(\mathbf{x}) \geq \left(\Gamma(d/2 + 1)\right)^{1/d} \pi^{-1/2} k^{-1/d}$, though it is unknown if this bound is sharp.

### 3.3 COMPARISON OF OPEN LOOP METHODS

In Figure 1 we plot the dispersion of the Sobol sequence, samples drawn uniformly at random, and samples drawn from a $k$-DPP, in one and two dimensions. To generate the $k$-DPP samples, we sequentially drew samples proportional to the (updated) posterior variance (using an RBF kernel, with $\sigma = \sqrt{2}/k$), as described in Section 2.2. When $d = 1$, the regular structure of the Sobol sequence causes it to have increasingly large plateaus, as there are many "holes" of the same size.[6] For example, the Sobol sequence has the same dispersion for $42 \leq k \leq 61$, and $84 \leq k \leq 125$. Samples drawn from a $k$-DPP appear to have the same asymptotic rate as the Sobol sequence, but they don't suffer from the plateaus. When $d = 2$, the $k$-DPP samples have lower average dispersion and lower variance.

One other natural surrogate of average optimization performance is to measure the distance from a fixed point, say $\frac{1}{2}\mathbf{1} = (\frac{1}{2}, \ldots, \frac{1}{2})$ or from the origin, to the nearest point in the length $k$ sequence. Our experiments (in Appendix D) on these metrics show the $k$-DPP samples bias samples to the corners of the space, which can be beneficial when the practitioner defined the search space with bounds that are too small.

Note, the low-discrepancy sequences are usually defined only for the $[0, 1]^d$ hypercrube, so for hyperparameter search which involves conditional hyperparameters (i.e. those with tree structure) they are not appropriate. In what follows, we study the $k$-DPP in more depth and how it performs on real-world hyperparameter tuning problems.

## 4 METHOD

We begin by reviewing DPPs and $k$-DPPs. Let $\mathcal{B}$ be a domain from which we would like to sample a finite subset. (In our use of DPPs, this is the set of hyperparameter assignments.) In general, $\mathcal{B}$ could be discrete or continuous; here we assume it is discrete with $N$ values, and we define $\mathcal{Y} = \{1, \ldots, N\}$ to be a a set which indexes $\mathcal{B}$ (this index set will be particularly useful in Algorithm 1). In Section 4.2 we address when $\mathcal{B}$ has continuous dimensions. A DPP defines a probability distribution over $2^{\mathcal{Y}}$ (all subsets of $\mathcal{Y}$) with the property that two elements of $\mathcal{Y}$ are more (less) likely to both be chosen the more dissimilar (similar) they are. Let random variable $\mathbf{Y}$ range over finite subsets of $\mathcal{Y}$.

There are several ways to define the parameters of a DPP. We focus on $\mathbf{L}$-ensembles, which define the probability that a specific subset is drawn (i.e., $P(\mathbf{Y} = \mathcal{A})$ for some $\mathcal{A} \subset \mathcal{Y}$) as:

$$P(\mathbf{Y} = \mathcal{A}) = \frac{\det(\mathbf{L}_{\mathcal{A}})}{\det(\mathbf{L} + I)}. \tag{1}$$

As shown in Kulesza et al. (2012), this definition of $\mathbf{L}$ admits a decomposition to terms representing the *quality* and *diversity* of the elements of $\mathcal{Y}$. For any $y_i, y_j \in \mathcal{Y}$, let:

$$\mathbf{L}_{i,j} = q_i q_j \mathcal{K}(\boldsymbol{\phi}_i, \boldsymbol{\phi}_j), \tag{2}$$

---

[4]Discrepancy is a global measure which depends on all points, while dispersion only depends on points near the largest "hole".

[5]In two dimensions a hexagonal tiling finds the optimal dispersion, but this is only valid when $k$ is divisible by the number of columns and rows in the tiling.

[6]By construction, each individual dimension of the $d$-dimensional Sobol sequence has these same plateaus.

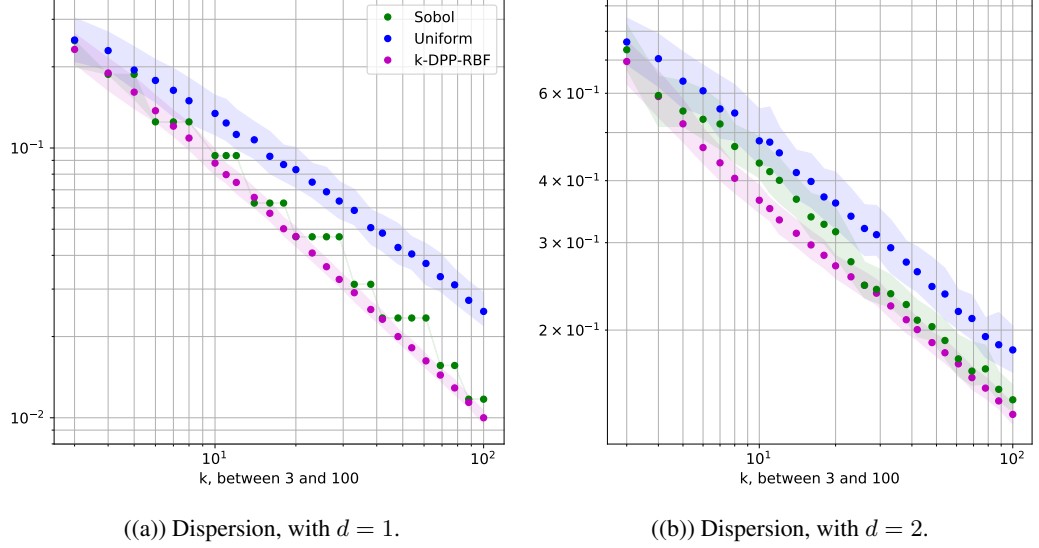

((a)) Dispersion, with $d = 1$.        ((b)) Dispersion, with $d = 2$.

Figure 1: Dispersion of the Sobol sequence, samples from a $k$-DPP, and uniform random samples (lower is better). These log-log plots show when $d = 1$ that Sobol suffers from regular plateaus of increasing size, while when $d = 2$ the $k$-DPP samples have lower average dispersion and lower variance.

where $q_i > 0$ is the quality of $y_i$, $\phi_i \in \mathbb{R}^d$ is a featurized representation of $y_i$, and $\mathcal{K} : \mathbb{R}^d \times \mathbb{R}^d \to [0, 1]$ is a similarity kernel (e.g. cosine distance). (We will discuss how to featurize hyperparameter settings in Section 4.3.)

Here, we fix all $q_i = 1$; in future work, closed loop methods might make use of $q_i$ to encode evidence about the quality of particular hyperparameter settings to adapt the DPP's distribution over time.

### 4.1 SAMPLING FROM A K-DPP

DPPs have support over all subsets of $\mathcal{Y}$, including $\emptyset$ and $\mathcal{Y}$ itself. In many practical settings, one may have a fixed budget that allows running the training algorithm $k$ times, so we require precisely $k$ elements of $\mathcal{Y}$ for evaluation. $k$-DPPs are distributions over subsets of $\mathcal{Y}$ of size $k$. Thus,

$$P(\mathbf{Y} = \mathcal{A} \mid |\mathbf{Y}| = k) = \frac{\det(\mathbf{L}_{\mathcal{A}})}{\sum_{\mathcal{A}' \subset \mathcal{Y}, |\mathcal{A}'| = k} \det(\mathbf{L}_{\mathcal{A}'})}. \tag{3}$$

Sampling from $k$-DPPs has been well-studied. When the base set $\mathcal{B}$ is a set of discrete items, exact sampling algorithms are known which run in $\mathcal{O}(Nk^3)$ Kulesza et al. (2012). When the base set is a continuous hyperrectangle, a recent exact sampling algorithm was introduced, based on a connection with Gaussian processes (GPs), which runs in $\mathcal{O}(dk^2 + k^3)$ Hennig & Garnett (2016). We are unaware of previous work which allows for sampling from $k$-DPPs defined over any other base sets.

### 4.2 SAMPLING K-DPPS DEFINED OVER ARBITRARY BASE SETS

Anari et al. (2016) present a Metropolis-Hastings algorithm (included here as Algorithm 1) which is a simple and fast alternative to the exact sampling procedures described above. However, it is restricted to discrete domains. We propose a generalization of the MCMC algorithm which preserves relevant computations while allowing sampling from any base set from which we can draw uniform samples, including those with discrete dimensions, continuous dimensions, some continuous and some discrete dimensions, or even (conditional) tree structures (Algorithm 2). To the best of our knowledge, this is the first algorithm which allows for sampling from a $k$-DPP defined over any space other than strictly continuous or strictly discrete, and thus the first algorithm to utilize the expressive capabilities of the posterior variance of a GP in these regimes.

---

**Algorithm 1** Drawing a sample from a discrete $k$-DPP (Anari et al., 2016)

---

**Input:** $\mathbf{L}$, a symmetric, $N \times N$ matrix where $\mathbf{L}_{i,j} = q_i q_j \mathcal{K}(\phi_i, \phi_j)$ which defines a DPP over a finite base set of items $\mathcal{B}$, and $\mathcal{Y} = \{1, \ldots, N\}$, where $\mathcal{Y}_i$ indexes a row or column of $\mathbf{L}$
**Output:** $\mathcal{B}_{\mathbf{Y}}$ (the points in $\mathcal{B}$ indexed by $\mathbf{Y}$)
1: Initialize $\mathbf{Y}$ to $k$ elements sampled from $\mathcal{Y}$ uniformly
2: **while** not mixed **do**
3:      uniformly sample $u \in \mathbf{Y}, v \in \mathcal{Y} \setminus \mathbf{Y}$
4:      set $\mathbf{Y}' = \mathbf{Y} \cup \{v\} \setminus \{u\}$
5:      $p \leftarrow \frac{1}{2} min(1, \frac{\det(\mathbf{L}_{\mathbf{Y}'})}{\det(\mathbf{L}_{\mathbf{Y}})})$
6:      with probability $p$: $\mathbf{Y} = \mathbf{Y}'$
7: Return $\mathcal{B}_{\mathbf{Y}}$

---

Algorithm 1 proceeds as follows: First, initialize a set $\mathbf{Y}$ with $k$ indices of $\mathbf{L}$, drawn uniformly. Then, at each iteration, sample two indices of $\mathbf{L}$ (one within and one outside of the set $\mathbf{Y}$), and with some probability replace the item in $\mathbf{Y}$ with the other.

When we have continuous dimensions in the base set, however, we can't define the matrix $\mathbf{L}$, so sampling indices from it is not possible. We propose Algorithm 2, which samples points directly from the base set $\mathcal{B}$ instead (assuming continuous dimensions are bounded), and computes only the principal minors of $\mathbf{L}$ needed for the relevant computations on the fly.

---

**Algorithm 2** Drawing a sample from a $k$-DPP defined over a space with continuous and discrete dimensions

---

**Input:** A base set $\mathcal{B}$ with some continuous and some discrete dimensions, a quality function $\mathbf{\Psi} : \mathbf{Y}_i \to q_i$, a feature function $\mathbf{\Phi} : \mathbf{Y}_i \to \phi_i$
**Output:** $\boldsymbol{\beta}$, a set of $k$ points in $\mathcal{B}$
1: Initialize $\boldsymbol{\beta}$ to $k$ points sampled from $\mathcal{B}$ uniformly
2: **while** not mixed **do**
3:      uniformly sample $u \in \boldsymbol{\beta}, v \in \mathcal{B} \setminus \boldsymbol{\beta}$
4:      set $\boldsymbol{\beta}' = \boldsymbol{\beta} \cup \{v\} \setminus \{u\}$
5:      compute the quality score for each item, $q_i = \mathbf{\Psi}(\boldsymbol{\beta}_i), \forall i$, and $q_i' = \mathbf{\Psi}(\boldsymbol{\beta}_i'), \forall i$
6:      construct $\mathbf{L}_{\boldsymbol{\beta}} = [q_i q_j \mathcal{K}(\mathbf{\Phi}(\boldsymbol{\beta}_i), \mathbf{\Phi}(\boldsymbol{\beta}_j))], \forall i, j$
7:      construct $\mathbf{L}_{\boldsymbol{\beta}'} = [q_i' q_j' \mathcal{K}(\mathbf{\Phi}(\boldsymbol{\beta}_i'), \mathbf{\Phi}(\boldsymbol{\beta}_j'))], \forall i, j$
8:      $p \leftarrow \frac{1}{2} min(1, \frac{\det(\mathbf{L}_{\boldsymbol{\beta}'})}{\det(\mathbf{L}_{\boldsymbol{\beta}})})$
9:      with probability $p$: $\boldsymbol{\beta} = \boldsymbol{\beta}'$
10: Return $\boldsymbol{\beta}$

---

Even in the case where the dimensions of $\mathcal{B}$ are discrete, Algorithm 2 requires less computation and space than Algorithm 1 (assuming the quality and similarity scores are stored once computed, and retrieved when needed). Previous analyses claimed that Algorithm 1 should mix after $\mathcal{O}(N \log(N))$ steps. There are $\mathcal{O}(N^2)$ computations required to compute the full matrix $L$, and at each iteration we will compute at most $O(k)$ new elements of $L$, so even in the worst case we will save space and computation whenever $k \log(N) < N$. In expectation, we will save significantly more.

### 4.3 Constructing $\mathbf{L}$ for hyperparameter optimization

Let $\phi_i$ be a feature vector for $y_i \in \mathcal{Y}$, a modular encoding of the attribute-value mapping assigning values to different hyperparameters, in which fixed segments of the vector are assigned to each hyperparameter attribute (e.g., the dropout rate, the choice of nonlinearity, etc.). For a hyperparameter that takes a numerical value in range $[h_{\min}, h_{\max}]$, we encode value $h$ using one dimension ($j$) of $\phi$ and project into the range $[0, 1]$:

$$\phi[j] = \frac{h - h_{\min}}{h_{\max} - h_{\min}} \tag{4}$$

This rescaling prevents hyperparameters with greater dynamic range from dominating the similarity calculations. A categorical-valued hyperparameter variable that takes $m$ values is given $m$ elements

of $\phi$ and a one-hot encoding. Ordinal-valued hyperparameters can be encoded using a unary encoding. (For example, an ordinal variable which can take three values would be encoded with [1,0,0], [1,1,0], and [1,1,1].) Additional information about the distance between the values can be incorporated, if it's available. In this work, we then compute similarity using an RBF kernel, $\mathcal{K} = \exp\left(-\frac{||\phi_i - \phi_j||^2}{2\sigma^2}\right)$, and hence label our approach $k$-DPP-RBF. Values for $\sigma^2$ lead to models with different properties; when $\sigma^2$ is small, points that are spread out interact little with one another, and when $\sigma^2$ is large, the increased repulsion between the points encourages them to be as far apart as possible.

### 4.4 Tree-structured hyperparameters

Many real-world hyperparameter search spaces are tree-structured. For example, the number of layers in a neural network is a hyperparameter, and each additional layer adds at least one new hyperparameter which ought to be tuned (the number of nodes in that layer). For a binary hyperparameter like whether or not to use regularization, we use a one-hot encoding. When this hyperparameter is "on," we set the associated regularization strength as above, and when it is "off" we set it to zero. Intuitively, with all other hyperparameter settings equal, this causes the off-setting to be closest to the least strong regularization. One can also treat higher-level design decisions as hyperparameters (Komer et al., 2014), such as whether to train a logistic regression classifier, a convolutional neural network, or a recurrent neural network. In this construction, the type of model would be a categorical variable (and thus get a one-hot encoding), and all child hyperparameters for an "off" model setting (such as the convergence tolerance for logistic regression, when training a recurrent neural network) would be set to zero.

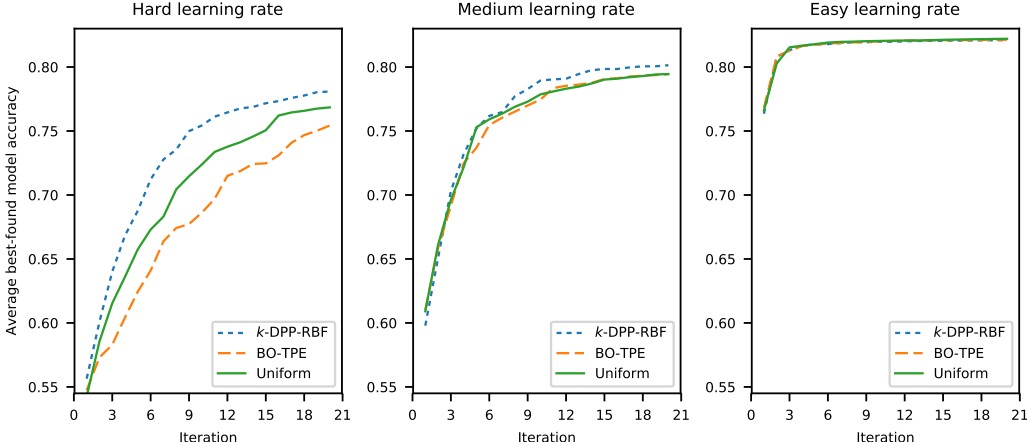

Figure 2: Average best-found model accuracy by iteration when training a convolutional neural network on three hyperparameter search spaces (defined in Section 5.1), averaged across 50 trials of hyperparameter optimization, with $k = 20$.

## 5 Hyperparameter Optimization Experiments

In this section we present our hyperparameter optimization experiments. Our experiments consider a setting where hyperparameters have a large effect on performance: a convolutional neural network for text classification (Kim, 2014). The task is binary sentiment analysis on the Stanford sentiment treebank (Socher et al., 2013). On this balanced dataset, random guessing leads to 50% accuracy. We use the CNN-non-static model from Kim (2014), with skip-gram (Mikolov et al., 2013) vectors. The model architecture consists of a convolutional layer, a max-over-time pooling layer, then a fully connected layer leading to a softmax. All $k$-DPP samples are drawn using Algorithm 2.

## 5.1 SIMPLE TREE-STRUCTURED SPACE

We begin with a search over three continuous hyperparameters and one binary hyperparameter, with a simple tree structure: the binary hyperparameter indicates whether or not the model will use $L_2$ regularization, and one of the continuous hyperparameters is the regularization strength. We assume a budget of $k = 20$ evaluations by training the convolutional neural net. $L_2$ regularization strengths in the range $[e^{-5}, e^{-1}]$ (or no regularization) and dropout rates in $[0.0, 0.7]$ are considered. We consider three increasingly "easy" ranges for the learning rate:

- Hard: $[e^{-5}, e^5]$, where the majority of the range leads to accuracy no better than chance.
- Medium: $[e^{-5}, e^{-1}]$, where half of the range leads to accuracy no better than chance.
- Easy: $[e^{-10}, e^{-3}]$, where the entire range leads to models that beat chance.

Figure 2 shows the accuracy (averaged over 50 runs) of the best model found after exploring 1, 2, ..., $k$ hyperparameter settings. We see that $k$-DPP-RBF finds better models with fewer iterations necessary than the other approaches, especially in the most difficult case. Figure 2 compares the sampling methods against a Bayesian optimization technique using a tree-structured Parzen estimator (BO-TPE; Bergstra et al., 2011). This technique evaluates points sequentially, allowing the model to choose the next point based on how well previous points performed (a closed loop approach). It is state-of-the-art on tree-structured search spaces (though its sequential nature limits parallelization). Surprisingly, we find it performs the worst, even though it takes advantage of additional information. We hypothesize that the exploration/exploitation tradeoff in BO-TPE causes it to commit to more local search before exploring the space fully, thus not finding hard-to-reach global optima.

Note that when considering points sampled uniformly or from a DPP, the order of the $k$ hyperparameter settings in one trial is arbitrary (though this is not the case with BO-TPE as it is an iterative algorithm). In all cases the variance of the best of the $k$ points is lower than when sampled uniformly, and the differences in the plots are all significant with $p < 0.01$.

## 5.2 OPTIMIZING WITHIN RANGES KNOWN TO BE GOOD

Zhang & Wallace (2015) analyzed the stability of convolutional neural networks for sentence classification with respect to a large set of hyperparameters, and found a set of six which they claimed had the largest impact: the number of kernels, the difference in size between the kernels, the size of each kernel, dropout, regularization strength, and the number of filters. We optimized over their prescribed "Stable" ranges for three open loop methods and one closed loop method; average accuracies with 95 percent confidence intervals from 50 trials of hyperparameter optimization are shown in Figure 3, across $k = 5, 10, 15, 20$ iterations. We find that even when optimizing over a space for which all values lead to good models, $k$-DPP-RBF outperforms the other methods.

Our experiments reveal that, while the hyperparameters proposed by Zhang & Wallace (2015), can have an effect, the learning rate, which they do not analyze, is at least as impactful.

## 5.3 WALL CLOCK TIME COMPARISON WITH SPEARMINT

Here we compare our approach against Spearmint (Snoek et al., 2012), perhaps the most popular Bayesian optimization package. Figure 4 shows wall clock time and accuracy for 25 runs on the "Stable" search space of four hyperparameter optimization approaches: $k$-DPP-RBF (with $k = 20$), batch Spearmint with 2 iterations of batch size 10, batch Spearmint with 10 iterations of batch size 2, and sequential Spearmint[7]. Each point in the plot is one hyperparameter assignment evaluation. The vertical lines represent how long, on average, it takes to find the best result in one run. We see that all evaluations for $k$-DPP-RBF finish quickly, while even the fastest batch method (2 batches of size 10) takes nearly twice as long on average to find a good result. The final average best-found accuracies are 82.61 for $k$-DPP-RBF, 82.65 for Spearmint with 2 batches of size 10, 82.7 for Spearmint with 10 batches of size 2, and 82.76 for sequential Spearmint. Thus, we find it takes on average more than ten times as long for sequential Spearmint to find its best solution, for a gain of only 0.15 percent accuracy.

---

[7]When in the fully parallel, open loop setting, Spearmint simply returns the Sobol sequence.

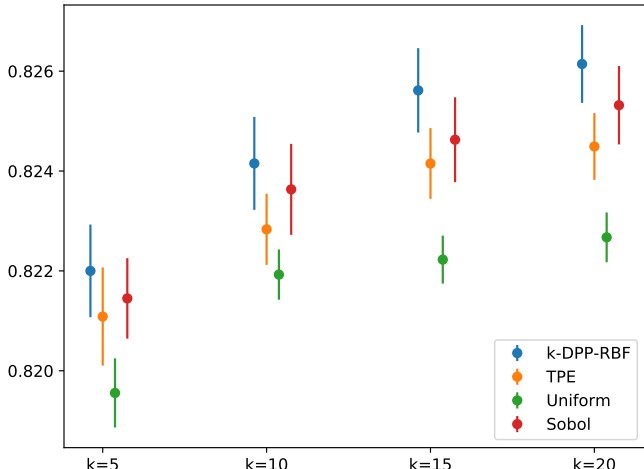

Figure 3: Average best-found model accuracy by iteration when training a convolutional neural network on the "Stable" search space (defined in Section 5.2), averaged across 50 trials of hyperparameter optimization, with $k = 5, 10, 15, 20$, with 95 percent confidence intervals. The $k$-DPP-RBF outperforms uniform sampling, TPE, and the Sobol sequence.

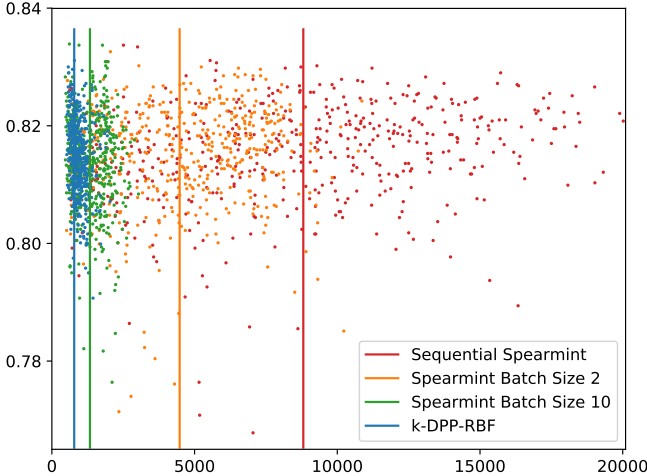

Figure 4: Wall clock time (in seconds, x-axis) for 25 hyperparameter trials of hyperparameter optimization (each with $k = 20$) on the "Stable" search space define in Section 5.2. The vertical lines represent the average time it takes too find the best hyperparameter assignment in a trial.

## 6 CONCLUSIONS

We have explored open loop hyperparameter optimization built on sampling from a $k$-DPP. We described how to define a $k$-DPP over hyperparameter search spaces, and showed that $k$-DPPs retain the attractive parallelization capabilities of random search. In synthetic experiments, we showed $k$-DPP samples perform well on a number of important metrics, even for large values of $k$. In hyperprameter optimization experiments, we see $k$-DPP-RBF outperform other open loop methods. Additionally, we see that sequential methods, even when using more than ten times as much wall clock time, gain less than 0.16 percent accuracy on a particular hyperparameter optimization problem. An open-source implementation of our method is available.[8]

---

[8]Anonymized URL; will be provided on publication.

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

## A   STAR DISCREPANCY

$$D_k(\mathbf{x}) = \sup_{u_1,\ldots,u_d \in [0,1]} \left| \mathcal{A}_k(\mathbf{x}, u_j) - \prod_{j=1}^{d} u_j \right|, \text{where}$$

$$\mathcal{A}_k(\mathbf{x}, u_j) = \frac{1}{k} \sum_{i=1}^{k} \mathbf{1} \left\{ x_i \in \prod_{j=1}^{d} [0, u_j) \right\}.$$

It is well-known that a sequence chosen uniformly at random from $[0,1]^d$ has an expected star discrepancy of at least $\sqrt{\frac{1}{k}}$ (and is no greater than $\sqrt{\frac{d \log(d)}{k}}$) (Shalev-Shwartz & Ben-David, 2014) whereas sequences are known to exist with star discrepancy less than $\frac{\log(k)^d}{k}$ (Sobol', 1967), where both bounds depend on absolute constants.

Comparing the star discrepancy of sampling uniformly and Sobol, the bounds suggest that as $d$ grows large relative to $k$, Sobol starts to suffer. Indeed, Bardenet & Hardy (2016) notes that the Sobol rate is not even valid until $k = \Omega(2^d)$ which motivates them to study a formulation of a DPP that has a star discrepancy between Sobol and random and holds for all $k$, small and large. They primarily approached this problem from a theoretical perspective, and didn't include experimental results. Their work, in part, motivates us to look at DPPs as a solution for hyperparameter optimization.

## B   KOKSMA-HLAWKA INEQUALITY

Let $\mathcal{B}$ be the $d$-dimensional unit cube, and let $f$ have bounded Hardy and Krause variation $Var_{HK}(f)$ on $\mathcal{B}$. Let $\mathbf{x} = (x_1, x_2, \ldots, x_k)$ be a set of points in $\mathcal{B}$ at which the function $f$ will be evaluated to approximate an integral. The Koksma-Hlawka inequality bounds the numerical integration error by the product of the star discrepancy and the variation:

$$\left| \frac{1}{k} \sum_{i=1}^{k} f(x_i) - \int_{\mathcal{B}} f(u) du \right| \leq Var_{HK}(f) D_k(\mathbf{x}).$$

We can see that for a given $f$, finding $\mathbf{x}$ with low star discrepancy can improve numerical integration approximations.

## C   ALGORITHM FOR COMPUTING DISPERSION

Find a (bounded) voronoi diagram over the search space for a point set $X_k$. For each vertex in the voronoi diagram, find the closest point in $X_k$. The dispersion is the max over these distances.

## D   DISTANCE TO THE CENTER AND THE ORIGIN

One natural surrogate of average optimization performance is to define a hyperparameter space on $[0,1]^d$ and measure the distance from a fixed point, say $\frac{1}{2}\mathbf{1} = (\frac{1}{2}, \ldots, \frac{1}{2})$, to the nearest point in the length $k$ sequence in the Euclidean norm squared: $\min_{i=1,\ldots,k} ||x_i - \frac{1}{2}\mathbf{1}||_2^2$. The Euclidean norm (squared) is motivated by a quadratic Taylor series approximation around the minimum of the hypothetical function we wish to minimize. In the first columns of Figure 5 we plot the smallest distance from the center $\frac{1}{2}\mathbf{1}$, as a function of the length of the sequence (in one dimension) for the Sobol sequence, uniform at random, and a DPP. We observe all methods appear comparable when it comes to distance to the center.

Acknowledging the fact that practitioners define the search space themselves more often than not, we realize that if the search space bounds are too small, the optimal solution often is found on the edge, or in a corner of the hypercube (as the true global optima are outside the space). Thus, in some situations it makes sense to *bias* the sequence towards the edges and the corners, the very opposite

of what low discrepancy sequences attempt to do. While Sobol and uniformly random sequences will not bias themselves towards the corners, a DPP does. This happens because points from a DPP are sampled according to how distant they are from the existing points; this tends to favor points in the corners. This same behavior of sampling in the corners is also very common for Bayesian optimization schemes, which is not surprise due to the known connections between sampling from a DPP and Gaussian processes (see Section 2.2). In the second column of Figure 5 we plot the distance to the origin which is just an arbitrarily chosen corner of hypercube. As expected, we observe that the DPP tends to outperform uniform at random and Sobol in this metric.

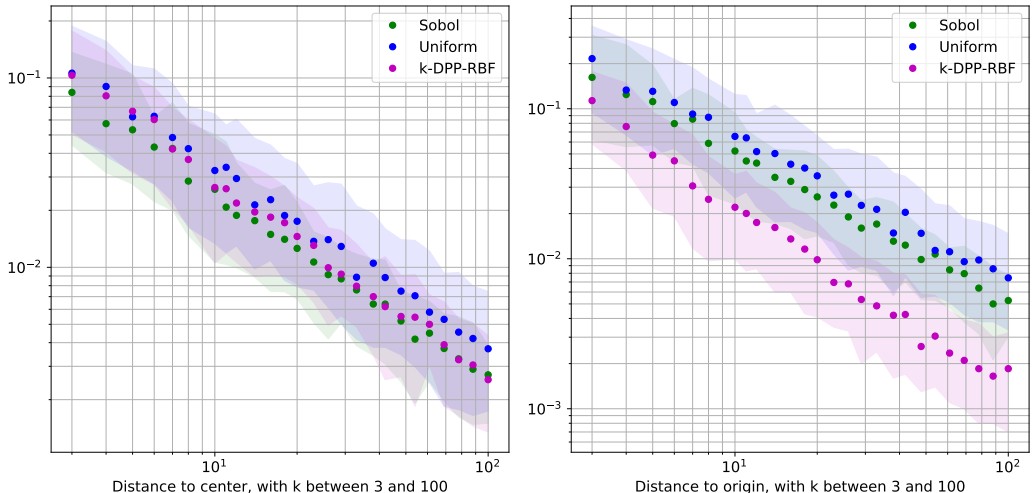

Figure 5: Comparison of the Sobol sequence, samples a from $k$-DPP, and uniform random for two metrics of interest. These log-log plots show uniform sampling and $k$-DPP-RBF performs comparably to the Sobol sequence in terms of distance to the center, but on another (distance to the origin) $k$-DPP-RBF samples outperform the Sobol sequence and uniform sampling.

