# OpenReview forum: "Open Loop Hyperparameter Optimization and Determinantal Point Processes"
_ICLR.cc/2019/Conference_

### Official Review · AnonReviewer3 · 2018-10-31
**recommending rejection because of lack of analyses and questionable novelty**

**Rating:** 5
**Confidence:** 3

**Review:**

The authors propose to use k-DPP to select a set of diverse parameters and use them to search for a good a hyperparameter setting.

This paper covers the related work nicely, with details on both closed loop and open loop methods. The rest of the paper are also clearly written. However, I have some concerns about the proposed method.
- It is not clear how to define the kernel, the feature function and the quality function for the proposed method. The choices of those seem to have a huge impact on the performance. How was those functions decided and how sensitive is the result to hyperparameters of those functions?
- If the search space is continuous, what is the mixing rate of Alg. 2? In practice, how is "mixed" decided? What exactly is the space and time complexity? I'm not sure where k log(N) comes from in page 7.
- Alg. 2 is a straight forward extension of Alg. 1, just with L not explicitly computed. I think it would have more novelty if some theoretical analyses can be shown on the mixing rate and how good this optimization algorithm is.

Other small things:
- citation format problems in, for example, Sec. 4.1. It should be \citep instead of \cite.
- it would be good to mention Figure 2 in the text first before showing it.

[Post rebuttal]
I would like to thank the authors for their clarifications. However, I am still concerned with the novelty. The absence of provable mixing rate is also a potential weakness. I think a clearer emphasis on the novelty, e.g. current algorithm with mixing rate analyses or more thorough empirical comparisons will make the paper stronger for resubmission.

---

> ### Author Response · Authors · 2018-11-20
> **Thank you for your review.**
>
> We thank AnonReviewer3 for their review, and address their points in order.
>
> We experimented with a number of different kernels, including cosine similarity (the most common approach used with K-DPPs), a kernel built using Levenshtein distance, and the RBF kernel. We found our results to be quite robust to the choice of the kernel -- we saw the K-DPP outperform the other approaches on our hyperparameter optimization experiments for all three. Similarly, we experimented with a number of different bandwidths for the RBF kernel, and found that as long as the bandwidth was large enough that the points interacted each other, it performed well. Our experiments focused on the presented feature function, as it was the most natural, but we expect any feature function which allows the points to be repulsed from one another (through the kernel) would behave similarly. In this work, we set the quality function to be 1, and have left learning the quality function to future work.
>
> If the search space is continuous, the mixing rate of Alg. 2 is not known. In practice, the MCMC algorithm is quite fast (a small fraction of the expense of training the models) so we ran the algorithm for 10x as long as the expected mixing rate if the space had been discrete, though our synthetic and real experimental results indicated that it was mixed significantly earlier.
> The k log(N) term appears when analyzing the difference between Algorithm 1 and Algorithm 2: computation and storage of the NxN kernel matrix L. In the discrete case, Algorithm 1 requires computing all of L, which has time and space complexity of O(N^2). Algorithm 2, instead of constructing L directly, only uses a submatrix of L computed on the fly. It runs for O(N log(N)) steps, and at each step computes and stores at most O(k) additional distances, leading to a total of O(Nk log(N)) time and space complexity (with a max of O(N^2) once it computes all of L). Therefore, Algorithm 2 has better complexity when O(Nk log(N)) < O(N^2), or when k log(N) < N. Otherwise, Algorithm 1 and 2 have the same time and space complexity. We will include a clarification in the paper, please let us know if this is still unclear.
>
> While we agree Algorithm 2 is a straightforward extension, it is an important one for the community. Other work that has used K-DPPs for hyperparameter optimization (Kathuria et al., 2016, Wang et al., 2017) has been restricted to non-tree structured domains, and has discretized continuous spaces to be discrete so they could use existing sampling algorithms, which we experimentally found to hurt performance. We introduce the ability to sample from more realistic hyperparameter spaces.
>
> Thank you for pointing out the small changes, they will be updated.
>
> The title of your review mentions worries about novelty, but as we mentioned in a reply to another review, we believe this approach (drawing samples from tree-structured, mixed discrete and continuous spaces in the open loop regime) and analysis (including dispersion calculation) are novel. We welcome further discussion, especially of more specific novelty concerns that may arise, and look forward to further suggestions on how to improve our paper or clarifications we can provide.

---

### Official Review · AnonReviewer2 · 2018-11-02
**interesting and clear idea on using DPPs for hyperparameter search**

**Rating:** 6
**Confidence:** 4

**Review:**

- This paper proposes an approach to get samples with high dispersion for hyperparameter optimisation.
- It theoretically motivates the use of Determinantal Point Processes in yielding such samples.
- Further, an iterative mixing algorithm is proposed to handle continuous and discrete sample space.
- Experiments on finding hyperparameter for sentence classification are presented. In terms of accuracy, it performs better than other open-loop methods. In comparison to closed-loop methods, it yields parameter settings with comparable performance but with gains in wall clock time.
- The distinction from close-loop approaches makes it easy to parallelise.


This paper is novel in its modelling of hyperparameter optimisation with DPP and the theoretical justification and experiments have been clearly presented. It would be interesting to explore the practicability of the method on more large-scale experiments on image related tasks.

---

> ### Author Response · Authors · 2018-11-20
> **How else can we improve?**
>
> We thank AnonReviewer2 for their thoughtful review.
>
> We have been convinced by the synthetic and experimental results (on two hyperparameter search spaces) of the efficacy and generality of our approach, and ask what experiments on an image dataset would contribute beyond the current experimental results?
>
> We look forward to further discussion, especially if there are any other points we can clarify, or additional ways you suggest we could improve our work.

---

### Official Review · AnonReviewer1 · 2018-11-05
**small number of hyperparameters, comparison with spearmint not strong enough**

**Rating:** 5
**Confidence:** 5

**Review:**

I reviewed the same paper last year. I am appending a few lines based on the changes made by authors.

The authors propose k-DPP as an open loop (oblivious to the evaluation of configurations) method for hyperparameter optimization and provide its empirical study and comparison with other methods such as grid search, uniform random search, low-discrepancy Sobol sequences, BO-TPE (Bayesian optimization using tree-structured Parzen estimator) by Bergstra et al. (2011). The k-DPP sampling algorithm and the concept of k-DPP-RBF over hyperparameters are not new, so the main contribution here is the empirical study.

The first experiment by the authors shows that k-DPP-RBF gives better star discrepancy than uniform random search while being comparable to low-discrepancy Sobol sequences in other metrics such as distance from the center or an arbitrary corner (Fig. 1).

The second experiment shows surprisingly that for the hard learning rate range, k-DPP-RBF performs better than uniform random search, and moreover, both of these outperform BO-TPE (Fig. 2, column 1).

The third experiment shows that on good or stable ranges, k-DPP-RBF and its discrete analog slightly outperform uniform random search and its discrete analog, respectively.

I have a few reservations. First, I do not find these outcomes very surprising or informative, except for the second experiment (Fig. 2, column 1). Second, their study only applies to a small number like 3-6 hyperparameters with a small k=20. The real challenge lies in scaling up to many hyperparameters or even k-DPP sampling for larger k. Third, the authors do not compare against some relevant, recent work, e.g., Springenberg et al. (http://aad.informatik.uni-freiburg.de/papers/16-NIPS-BOHamiANN.pdf) and Snoek et al. (https://arxiv.org/pdf/1502.05700.pdf) that is essential for this kind of empirical study.

COMMENTS ON THE CHANGES SINCE THE LAST YEAR

I am not convinced by the comparison with Spearmint added by the authors since the previous version. It is unclear to me if the comparison of wall clock time and accuracy holds for larger number of hyperparameters or against Spearmint with more parallelization.

In addition the authors do not compare against more recent work, e.g.,

@INPROCEEDINGS{falkner-bayesopt17,
 author    = {S. Falkner and A. Klein and F. Hutter},
 title     = {Combining Hyperband and Bayesian Optimization},
 booktitle = {NIPS 2017 Bayesian Optimization Workshop},
 year      = {2017},
 month     = dec,
}

@InProceedings{falkner-icml-18,
  title =        {{BOHB}: Robust and Efficient Hyperparameter Optimization at Scale},
  author =       {Falkner, Stefan and Klein, Aaron and Hutter, Frank},
  booktitle =    {Proceedings of the 35th International Conference on Machine Learning (ICML 2018)},
  pages =        {1436--1445},
  year =         {2018},
  month =        jul,
}

---

> ### Author Response · Authors · 2018-11-20
> **Thank you for both rounds of review you have provided for our paper.**
>
> We greatly appreciate both rounds of review that you've provided for our paper. We took the first review quite seriously, and it seems that you missed some of the changes we made in response to your review and others.  Specifically, addressing items that have changed that were mentioned in your first review:
> The first experiment actually shows dispersion, not discrepancy. While we include star discrepancy in the appendix (in Figure 5, not Figure 1), we argue in section 3 that dispersion is a better measure of the quality of a point set for optimization (though discrepancy has been the tool of choice for previous work). We include a theorem which bounds the optimization error with dispersion, a connection which discrepancy lacks, and show our approach outperforms the Sobol sequence and uniform sampling. We believe this connection will encourage future work to move away from evaluating open loop methods with discrepancy, and to use dispersion instead.
> The third experiment does not address discretization error. Instead, it is another hyperparameter optimization experiment on a different search space, again showing that samples from a K-DPP outperform uniform samples, the Sobol sequence, and BO-TPE.
>
> We believe our method for defining a K-DPP over tree-structured, mixed discrete-continuous spaces is novel, as is the sampling algorithm we introduce in Algorithm 2. We know of no other approach that can draw K-DPP samples from tree-structured, mixed discrete-continuous domains. Previous work using K-DPPs for hyperparameter optimization (Kathuria et al., 2016, Wang et al., 2017) discretize continuous domains, then use a known algorithm to sample from a discrete (non-tree structured) base set. Your first review mentions a plot which showed the error from discretizing the search space, empirically motivating this contribution.
>
> To address the listed reservations: we do find it surprising that samples from a K-DPP match or outperform the Sobol sequence in our synthetic measures, as the Sobol sequence was designed specifically to perform well. Additionally, it has become perhaps the most frequently used approach (e.g. without function evaluation results, Spearmint returns the Sobol sequence, while our results indicate that it should return K-DPP samples instead).
>
> We agree that scaling into large K and D is important, but that isn't the focus of this work, and there is a large body of work on improving space and time complexity of GPs which is directly applicable to our approach.
>
> When considering which pieces of recent work to compare against, we emphasize that our work is not trying to answer the question, "Is active learning helpful?" by comparing active learning approaches against our open loop approach; instead, we focus on comparing against other non-active learning approaches. For example, Hyperband starts by uniformly sampling K hyperparameter assignments, then (partially) training and evaluating models with those assignments. This work does not advocate replacing Hyperband with a single draw from a K-DPP (i.e., replacing an active learning strategy with a non-active learning strategy), but it does argue that the uniform sampling step in Hyperband be replaced with a draw from a K-DPP (replacing an open loop strategy with another, better one). All of your suggested comparisons are against active learning approaches.
>
> We do include experiments comparing against Spearmint (an active learning approach), though this is meant to illustrate the large cost in optimization time that active learning entails. We appreciate the suggestion to compare against Spearmint with more parallelization, but note that in our experiments we compared against the most parallel possible Spearmint configuration (as well as a number of others). Any active learning strategy (excluding Hyperband-style evaluations of partially trained models) will take at least twice as long in expectation as a fully-parallel non-active learning strategy like a K-DPP to train and evaluate models with a set of hyperparameter assignments, so we expect the results in Figure 4 to hold for any number of hyperparameters.
>
> Thank you again for your review, we look forward to further discussion.

---

### Meta-Review · Area_Chair1 · 2018-12-16
**A well written, interesting paper on designing experiments for hyperparameter optimization using DPPs, but with lingering concerns over novelty and experiments.**

**Confidence:** 5
**Recommendation:** Reject

**Metareview:**

This is a very clearly written, well composed paper that does a good job of placing the proposed contribution in the scope of hyperparameter optimization techniques.  This paper certainly appears to have been improved over the version submitted to the previous ICLR.  In particular, the writing is much clearer and easy to follow and the methodology and experiments have been improved.  The ideas are well motivated and it's exciting to see that sampling from a k-DPP can give better low discrepancy sequences than e.g. Sobol.  However,  the reviewers still seem to have two major concerns, namely novelty of the approach (DPPs have been used for Bayesian optimization before) and the empirical evaluation.

Empirical evaluation:  As Reviewer1 notes, there are much more recent approaches for Bayesian optimization that have improved significantly over the TPE method, also for conditional parameters.  There are also more recent approaches proposing variants of random search such as hyperband.

Novelty:  There is some work on using determinantal point processes for Bayesian optimization and related work in optimal experimental design.  Optimal design has a significant amount of literature dedicated to designing a set of experiments according to the determinant of their covariance matrix - i.e. D-Optimal Design.  This work may add some interesting contributions to that literature, including fast sampling from k-DPPs, etc.  It would be useful, however, to add some discussion of that literature in the paper.  Jegelka and Sra's tutorial at NeurIPS on negative dependence had a nice overview of some of this literature.

Unfortunately, two of the three reviewers thought the paper was just below the borderline and none of the reviewers were willing to champion it.  There are very promising and interesting ideas in the paper, however, that have a lot of potential.  In the opinion of the AC, one of the most powerful aspects of DPPs over e.g. low discrepancy sequences, random search, etc.  is the ability to learn a distance over a space under which samples will be diverse.  This can make a search *much* more efficient since (as the authors note when discussing random search vs. grid search) the DPP can sample more densely in areas and dimensions that have higher sensitivity.  It would be exciting to learn kernels specifically for hyperparameter optimization problems (e.g. a kernel specifically for learning rates that can capture e.g. logarithmic scaling).  Taking the objective into account through the quality score, as proposed for future work, also seems very sensible and could significantly improve results as well.